

# A regularized stochastic configuration network based on weighted mean of vectors for regression

Yang Wang[1], Tao Zhou[1], Guanci Yang[2], Chenglong Zhang[3] and Shaobo Li[1]

[1] State Key Laboratory of Public Big Data, Guizhou University, Guiyang, Guizhou, China
[2] Key Laboratory of Advanced Manufacturing Technology of the Ministry of Education, Guizhou University, Guiyang, Guizhou, China
[3] School of Computer Science and Technology, China University of Mining and Technology, Xuzhou, Jiangsu, China

## ABSTRACT

The stochastic configuration network (SCN) randomly configures the input weights and biases of hidden layers under a set of inequality constraints to guarantee its universal approximation property. The SCN has demonstrated great potential for fast and efficient data modeling. However, the prediction accuracy and convergence rate of SCN are frequently impacted by the parameter settings of the model. The weighted mean of vectors (INFO) is an innovative swarm intelligence optimization algorithm, with an optimization procedure consisting of three phases: updating rule, vector combining, and a local search. This article aimed at establishing a new regularized SCN based on the weighted mean of vectors (RSCN-INFO) to optimize its parameter selection and network structure. The regularization term that combines the ridge method with the residual error feedback was introduced into the objective function in order to dynamically adjust the training parameters. Meanwhile, INFO was employed to automatically explore an appropriate four-dimensional parameter vector for RSCN. The selected parameters may lead to a compact network architecture with a faster reduction of the network residual error. Simulation results over some benchmark datasets demonstrated that the proposed RSCN-INFO showed superior performance with respect to parameter setting, fast convergence, and network compactness compared with other contrast algorithms.

## INTRODUCTION

Neural networks have shown superiority over data modeling because of their powerful representation learning ability to learn patterns with multiple levels of abstraction that make sense to the data (*Bengio, Courville & Vincent, 2013*). However, the gradient-based iterative training process of neural networks is time-consuming and computationally intensive (*Wang & Li, 2017b*). Feed-forward neural networks (FNNs) with random parameters have drawn widespread attention due to their faster training speed and lower computational cost (*Scardapane & Wang, 2017*). *Igelnik & Pao (1995)* found that any

Corresponding author
Yang Wang, yangwang@gzu.edu.cn

continuous function can be approximated by a random vector functional link (RVFL) network with probability one under appropriate parameters. The hidden parameters of RVFL were assigned randomly in a preset scope and the output weights were calculated based on the least squares method (*Cao et al., 2021*). However, determining the preset scope of randomized learning models is challenging, and the widely used scope of random parameters (*e.g.*, [−1,1]) is not always feasible (*Li & Wang, 2017*).

To resolve the infeasibility of using RVFL networks for data modeling with a fixed scope (*i.e.*, [−1,1]), *Wang & Li (2017b)* proposed a novel randomized learning framework, termed SCN. The hidden parameters (input weights and biases) of SCN are randomly assigned under a supervisory mechanism and adaptively select their ranges, which indicate prominent merits on human intervention of network structure, range adaptation of hidden layer parameters, and sound generalization (*Dai et al., 2019a*).

Many efforts have been made to enhance the performance of SCN since it was developed in 2017. SCN with kernel density estimation (RSC-KDE) and maximum correntropy criterion (RSC-MCC) were presented to weaken the negative influences of noise and outliers, respectively, on modeling performance (*Wang & Li, 2017a*; *Li, Huang & Wang, 2019*). *Zhu et al. (2019)* delved deeper into the inequality constraint used in SCN and presented two new inequalities to increase the probability of satisfying the constraint condition. As deep neural networks (DNNs) with multiple levels of feature extraction can learn more abstract representations of the data, a deep version of SCN (DeepSCN) with multi-hidden layer network structure was proposed by *Wang & Li (2018)*. For image data analysis with matrix inputs, a two-dimensional version of SCN (2DSCN) was proposed by *Li & Wang (2019)*. For SCN ensembles, *Wang & Cui (2017)* adopted the negative correlation learning (NCL) ensemble learning technique to reduce the covariance among the base SCN for large-scale data analysis. *Huang, Li & Wang (2021)* designed a novel indicator that contained some key factors to explore appropriate base learner models from a set of SCN to generate an effective ensemble model. *Zhang et al. (2021)* developed a parallel SCN (PSCN) by introducing the beetle antennae search (BAS) optimization algorithm and fuzzy evidence theory for large-scale data regression. For finding the optimal parameter settings, *Zhang & Ding (2021)* utilized the chaotic sparrow search algorithm to optimize the contractive factor $r$ in the inequality and the scale factor $\lambda$ of random parameters to enhance the effectiveness of SCN. In addition, various extensions of SCN were applied to data modeling in real-world applications, such as molten iron quality (MIQ) modeling in blast furnace ironmaking (BFI) (*Xie & Zhou, 2020*), particle size estimation of hematite grinding process (*Dai et al., 2019b*), traffic state prediction across geo-distributed data centers of the China Southern Power Grid (CSG) (*Huang, Huang & Wang, 2019*), and prediction of asphaltene and total nitrogen in crude oil (*Lu & Ding, 2019*; *Lu et al., 2020*).

SCN starts with a small-sized network structure and gradually adds new hidden nodes into the network until the residual error of SCN is smaller than the tolerance threshold.

With the increasing number of hidden nodes, the constructive SCN model is prone to overfitting and thus poor performance (*Wang et al., 2021*). Meanwhile, the performance of SCN is frequently impacted by the parameter settings of the model, such as $\lambda$ (the scale factor of weights and bias) and $r$ (the contractive factor in the inequality). Seeking better model parameters is vital for SCN. It is well known that the $L_2$ regularization technique, which adds the "squared magnitude" of the coefficient to the loss function, can prevent the problem of overfitting effectively. In the famous Residual Network (ResNet), *He et al. (2016)* let the stacked nonlinear layers fit a residual mapping of $F(x) := H(x) - x$. Inspired by the idea of residual learning in ResNet, we used the current network residual error feedback to dynamically adjust the parameters of SCN.

Therefore, the objective of this study was to automatically obtain better parameters for SCN and get a more compact architecture. First, the $L_2$ regularization item combined with network residual error was introduced to improve the generalization performance of SCN. In addition, a regularized SCN based on INFO was developed to optimize the parameter selection of SCN. INFO is a relatively new swarm intelligence optimization method published in 2022. Updating rule, vector combining, and a local search were the three core phases of INFO (*Ahmadianfar et al., 2022*). It is a promising tool for the parameter optimization of the regularized stochastic configuration network (RSCN). To summarize, the key contributions of this article are as follows:

- Introduce the regularization term that combines the ridge method with the network residual error into the objective function to dynamically adjust the training parameters of SCN.
- Optimize the scope setting of the input weights and biases $\lambda$, contractive factor $r$ in the inequality, regularization coefficient $\eta$, and positive scale factor $\gamma$ of feedback residual error of RSCN by INFO, which in turn achieves a better RSCN model with respect to fast convergence and structure compactness.
- Illustrate the merits of RSCN-INFO on one function approximation and three benchmark regression datasets. The evaluation results justify the effectiveness of the proposed RSCN-INFO.

## PRELIMINARIES

This section briefly reviews the classical SCN framework and the newer INFO algorithm.

### SCN

SCN is a novel randomized incremental learner framework with a supervisory mechanism. The universal approximation property of SCN is guaranteed by its innovative inequality constraint. The network structure of SCN is depicted in Fig. 1.

Let $\Gamma := \{g_1, g_2, g_3, \ldots\}$ be a set of real-valued functions, span($\Gamma$) denotes a function space spanned by $\Gamma$, and $L_2(D)$ represents the space of all Lebesgue measurable functions

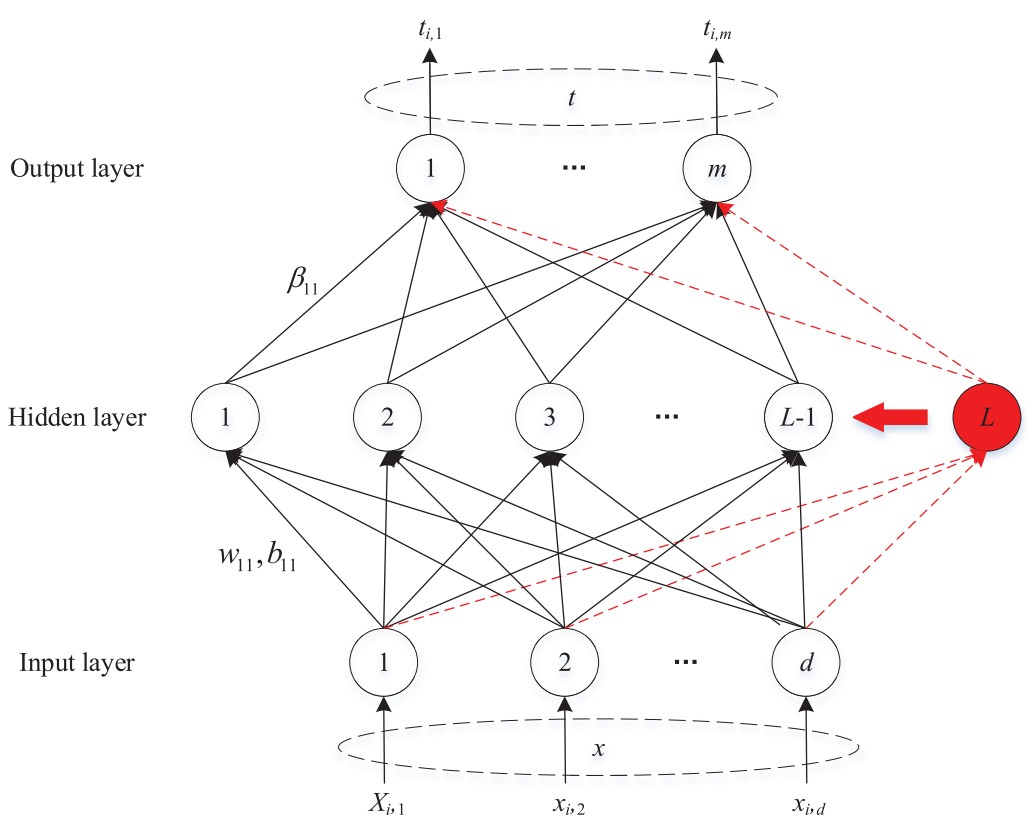

**Figure 1 The network structure of SCN.** 

$f = [f_1, f_2, \cdots, f_m] : \mathbb{R}^d \rightarrow \mathbb{R}^m$ on a set $D \subset \mathbb{R}^d$. $f_{L-1}(x) = \sum\limits_{j}^{L-1} \beta_j g_j(w_j^T x + b_j)$ represents

the output of a single layer feed-forward network with $L-1$ hidden nodes, where $w_j$ and $b_j$ denote the parameters of the $j$th hidden node, and $g_j(\cdot)$ is the activation function of the $j$th hidden node. The inner product of $\theta = [\theta_1, \theta_2, \ldots, \theta_m] : \mathbb{R}^d \rightarrow \mathbb{R}^m$ and $f$ is:

$$<f, \theta> := \sum_{q=1}^{m} <f_q, \theta_q> = \sum_{q=1}^{m} \int_D f_q(x) \theta_q(x) dx. \tag{1}$$

Given a training dataset $(x_i, t_i)_{i=1}^N$, $x_i = [x_{i,1}, \ldots, x_{i,d}] \in \mathbb{R}^d$ and $t_i = [t_{i,1}, \ldots, t_{i,m}] \in \mathbb{R}^m$, suppose that an established SCN model contains $L-1$ hidden nodes. The network residual error is:

$$E_{L-1}(x) = f(x) - f_{L-1}(x) = [E_{L-1,1}(x), E_{L-1,2}(x), \ldots, E_{L-1,m}(x)] \in \mathbb{R}^{N \times m},$$
$$E_{L-1,1}(x) = [E_{L-1,1}(x_1), E_{L-1,1}(x_2), \ldots, E_{L-1,1}(x_N)]^T,$$
$$E_{L-1,2}(x) = [E_{L-1,2}(x_1), E_{L-1,2}(x_2), \ldots, E_{L-1,2}(x_N)]^T,$$
$$\vdots \tag{2}$$
$$E_{L-1,m}(x) = [E_{L-1,m}(x_1), E_{L-1,m}(x_2), \ldots, E_{L-1,m}(x_N)]^T.$$

where $f$ is the given target function, and $f_{L-1}$ represents the output of the network with $L-1$ hidden nodes. The residual error of SCN gradually decreased as the number of hidden neurons increased. If the value of $E_{L-1}$ was larger than the tolerance threshold $\tau$, SCN added a new hidden node into the network until $E_L$ was smaller than $\tau$. The parameters of the added hidden node were assigned randomly under a set of inequalities.

$$< E_{L-1,q}, g_L >^2 \geq b_g^2(1 - r - \mu_L)||e_{L-1,q}||^2, q = 1, 2, \ldots, m, \tag{3}$$

where $< E_{L-1,q}, g_L >$ is the inner product of $E_{L-1,q}$ and $g_L$, $\{\mu_L\}$ represents a nonnegative real number sequence with $lim_{L \to +\infty} \mu_L = 0$, and $\mu_L \leq (1 - r)$. $g$ indicates a non-linear activation function. $\forall g \in \Gamma$ (span($\Gamma$) is dense in $L_2$ space), $0 < \| g \| < b_g$ ($b_g \in \mathbb{R}^+$). $r$ determines the strictness of the inequality constraint, and $0 < r < 1$. The output weights were evaluated by:

$$\beta^* = \arg \min_{\beta}||H_L\beta - T||_F^2 = H_L^\dagger T, \tag{4}$$

where $H_L^\dagger$ represents the Moore-Penrose inverse of $H_L$ and $H_L$ is the output matrix of the hidden layer. Readers may refer to *Wang & Li (2017b)* for more details on the SCN framework and associated algorithms.

## INFO

INFO is a new population-based optimization algorithm that employs updating rule, vector combining, and local search to move the population's position in $D$ dimensional search domains. Given a population with $N_P$ vectors, $X_{l,j}^g = \{x_{l,1}^g, x_{l,2}^g, \ldots, x_{l,D}^g\}, l = 1, 2, \ldots, N_P$.

- Updating rule

INFO randomly selected three differential vectors ($a1 \neq a2 \neq a3$) to calculate the weighted mean of vectors. To increase the diversity of the population, the best, better, and worst solutions were employed to define the *MeanRule* (mean-based rule).

$$MeanRule = k \times WM1_l^g + (1 - k) \times WM2_l^g. \tag{5}$$

$$WM1_l^g = \alpha \times \frac{w_1(x_{a1} - x_{a2}) + w_2(x_{a1} - x_{a3}) + w_3(x_{a2} - x_{a3})}{w_1 + w_2 + w_3 + \varepsilon} + \varepsilon \times rand, \tag{6}$$

$$l = 1, 2, \ldots, N_p,$$

where

$$w_1 = \cos((f(x_{a1}) - f(x_{a2})) + \pi) \times exp\left(-\frac{f(x_{a1}) - f(x_{a2})}{\omega}\right). \tag{7}$$

$$w_2 = \cos((f(x_{a1}) - f(x_{a3})) + \pi) \times exp\left(-\frac{f(x_{a1}) - f(x_{a3})}{\omega}\right). \tag{8}$$

$$w_3 = \cos((f(x_{a2}) - f(x_{a3})) + \pi) \times exp\left(-\frac{f(x_{a2}) - f(x_{a3})}{\omega}\right). \tag{9}$$

$$\omega = max(f(x_{a1}), f(x_{a2}), f(x_{a3})). \tag{10}$$

$$WM2_l^g = \alpha \times \frac{w_1(x_{bs} - x_{bt}) + w_2(x_{bs} - x_{ws}) + w_3(x_{bt} - x_{ws})}{w_1 + w_2 + w_3 + \varepsilon} + \varepsilon \times rand, \tag{11}$$

$$l = 1, 2, \ldots, N_P,$$

where

$$w_1 = \cos((f(x_{bs}) - f(x_{bt})) + \pi) \times exp\left(-\frac{f(x_{bs}) - f(x_{bt})}{\omega}\right). \tag{12}$$

$$w_2 = \cos((f(x_{bs}) - f(x_{ws})) + \pi) \times exp\left(-\frac{f(x_{bs}) - f(x_{ws})}{\omega}\right). \tag{13}$$

$$w_3 = \cos((f(x_{bt}) - f(x_{ws})) + \pi) \times exp\left(-\frac{f(x_{bt}) - f(x_{ws})}{\omega}\right). \tag{14}$$

$$\omega = f(x_{ws}). \tag{15}$$

The weighted mean of vectors was used to generate two new vectors.

$$\begin{cases} \begin{cases} z1_l^g = x_l^g + \sigma \times MeanRule + randn \times \frac{(x_{bs} - x_{a1}^g)}{(f(x_{bs}) - f(x_{a1}^g) + 1)}, \\ z2_l^g = x_{bs} + \sigma \times MeanRule + randn \times \frac{(x_{a1}^g - x_{a2}^g)}{(f(x_{a1}^g) - f(x_{a2}^g) + 1)}, \end{cases} & rand < 0.5, \\ \begin{cases} z1_l^g = x_a^g + \sigma \times MeanRule + randn \times \frac{(x_{a2}^g - x_{a3}^g)}{(f(x_{a2}^g) - f(x_{a3}^g) + 1)}, \\ z2_l^g = x_{bt} + \sigma \times MeanRule + randn \times \frac{(x_{a1}^g - x_{a2}^g)}{(f(x_{a1}^g) - f(x_{a2}^g) + 1)}, \end{cases} & rand \geq 0.5, \end{cases} \tag{16}$$

where $f(x)$ was defined as the objective function, three different integers $(a1, a2, a3)$ were randomly chosen from $[1, N_P]$, $z1_l^g$ and $z2_l^g$ were two new vectors, and $\sigma$ was the scaling rate of a vector.

● Vector combining

The two new vectors $z1_l^g$ and $z2_l^g$ were combined with vector $x_l^g$ to generate a new vector $\mu_l^g$.

$$\begin{cases} \begin{cases} \mu_l^g = z1_l^g + \mu.z1_l^g - z2_l^g, rand < 0.5, \\ \mu_l^g = z2_l^g + \mu.z1_l^g - z2_l^g, rand \geq 0.5, \end{cases} & rand < 0.5, \\ \mu_l^g = x_l^g, & rand \geq 0.5, \end{cases} \tag{17}$$

where $\mu_l^g$ was the composite vector of the $g$th generation.

● Local search

The local search operator used the global position $(x_{best}^g)$ and the MeanRule to help INFO convergence to global optima.

$$\begin{cases} \mu_l^g = x_{bs} + randn \times (MeanRule + randn \times (x_{bs}^g - x_{a1}^g)), & rand < 0.5, \\ \mu_l^g = x_{rnd} + randn \times (MeanRule + randn \times (v_1 \times x_{bs} - v_2 \times x_{rnd})), & rand \geq 0.5, \end{cases} \tag{18}$$

in which

$$x_{rnd} = \phi \times x_{avg} + (1 - \phi) \times (\phi \times x_{bt} + (1 - \phi) \times x_{bs}). \tag{19}$$

$$x_{avg} = \frac{(x_a + x_b + x_c)}{3}. \tag{20}$$

where $rand$ and $\phi$ were two random values within [0,1] and (0,1) respectively. The random value $v_1$ and $v_2$ increased the best position's influence on the vector. INFO updated the best vector ($x_{best}$) and returned $x_{best}^g$ as the final solution. For more details about the INFO algorithm, refer to *Ahmadianfar et al. (2022)*.

## RSCN-INFO

### RSCN

Given a training dataset $(x_i, t_i)_{i=1}^N$, the objective function of the SCN with $L_2$-norm penalty term could be expressed as:

$$min : J = \frac{1}{2} ||\beta||^2 + \frac{\eta}{2} \sum_{i=1}^N E_i^2, s.t : h(x_i)\beta = t_i - E_i, \forall i, \tag{21}$$

where $h(x_i)$ stands for the hidden output for the input $x_i$, $\eta$ is a non-negative real number, and the regularization coefficient $\eta$ balances the residual error $\left( \sum_{i=1}^N E_i^2 \right)$ and norm of the output weights ($||\beta||^2$).

SCN added new hidden neuron $\beta_L, g_L$, incrementally leading to $f_L = f_{L-1} + \beta_L g_L$. After dynamically adjusting the output weights during the training process, the current residual error was added to the $L_2$ regularization term. After adding the $L$th new hidden node into an established SCN model with $L - 1$ hidden nodes, a new objective function was introduced.

$$\begin{aligned} f(\beta_L) &= \frac{1}{2} [\beta \quad \beta_L] \begin{bmatrix} \beta \\ \beta_L \end{bmatrix} + \frac{\eta}{2} ||E_L||^2 \\ &= \frac{1}{2} ||\beta||^2 + \frac{1}{2} ||\beta_L||^2 + \frac{\eta}{2} ||E_{L-1} - \beta_L \left( g_L + \frac{||g_L||^2}{\gamma} E_{L-1} \right)||^2. \end{aligned} \tag{22}$$

where $\gamma$ is the positive scale factor of feedback residual error. The derivative of function Eq. (22) with respect to $\beta_L$ is:

$$\frac{\partial f(\beta_L)}{\partial \beta_L} = \beta_L - \eta \left\langle E_{L-1}, g_L + \frac{||g_L||^2}{\gamma} E_{L-1} \right\rangle + \eta \beta_L ||g_L + \frac{||g_L||^2}{\gamma} E_{L-1}||^2. \tag{23}$$

Letting Eq. (23) be equal to 0, The output weights of the $L$-th hidden node was obtained by:

$$\beta_L = \frac{\langle E_{L-1}, g_L + \frac{\|g_L\|^2}{\gamma} E_{L-1}\rangle}{\|g_L + \frac{\|g_L\|^2}{\gamma} E_{L-1}\|^2 + \frac{1}{\eta}}. \tag{24}$$

**Theorem 1.** Assume that $span(\Gamma)$ is dense in $L_2$ space. Given $0 < r < 1$, $0 < \eta$, $0 < \gamma$, and a nonnegative real number sequence $\mu_L$, with $\mu_L \geq 1 - r$ and $\lim_{L \to +\infty} \mu_L = 0$. $\forall g \in \Gamma$, $0 < \|g\| < b_g$ for some $b_g \in \mathbb{R}^+$. For $L = 1, 2, \ldots$ and $q = 1, 2, \ldots, m$, the random basis function $g_L$ is generated by Eq. (25), and the output weights of the $L$th hidden neuron are obtained by Eq. (26). Then, the SCN has $\lim_{L \to +\infty} \|f - f_L\| = 0$.

$$\frac{\langle E_{L-1,q}, g_L + \frac{\|g_L\|^2}{\gamma} E_{L-1,q}\rangle^2}{\left(\|g_L + \frac{\|g_L\|^2}{\gamma} E_{L-1,q}\|^2 + \frac{1}{\eta}\right)^2 / \left(\|g_L + \frac{\|g_L\|^2}{\gamma} E_{L-1,q}\|^2 + \frac{2}{\eta}\right)} \geq (1 - r - \mu_L)\|E_{L-1,q}\|^2, \tag{25}$$

$q = 1, 2, \ldots, m$.

$$\beta_{L,q} = \frac{\langle E_{L-1,q}, g_L + \frac{\|g_L\|^2}{\gamma} E_{L-1,q}\rangle}{\|g_L + \frac{\|g_L\|^2}{\gamma} E_{L-1,q}\|^2 + \frac{1}{\eta}}, q = 1, 2, \ldots, m. \tag{26}$$

*Proof.* First, the monotonically decreasing property of $\|E_L\|$ will be proved.

$$\|E_L\|^2 - \|E_{L-1}\|^2 =$$

$$\sum_{q=1}^{m} \left( \left\langle \left( E_{L-1,q} - \beta_L \left( g_L + \frac{\|g_L\|^2}{\gamma} E_{L-1,q}\right) \right), E_{L-1,q} - \beta_L \left( g_L + \frac{\|g_L\|^2}{\gamma} E_{L-1,q}\right) \right\rangle - \left\langle E_{L-1,q}, E_{L-1,q}\right\rangle \right)$$

$$= \sum_{q=1}^{m} \left( -2 \left\langle E_{L-1,q}, \beta_L \left( g_L + \frac{\|g_L\|^2}{\gamma} E_{L-1,q}\right) \right\rangle + \left\langle \beta_L \left( g_L + \frac{\|g_L\|^2}{\gamma} E_{L-1,q}\right), \beta_L \left( g_L + \frac{\|g_L\|^2}{\gamma} E_{L-1,q}\right) \right\rangle \right)$$

$$= \sum_{q=1}^{m} \left( -2 \frac{\left\langle E_{L-1,q}, \beta_L \left( g_L + \frac{\|g_L\|^2}{\gamma} E_{L-1,q}\right)\right\rangle^2}{\|g_L + \frac{\|g_L\|^2}{\gamma} E_{L-1,q}\|^2 + \frac{1}{\eta}} + \frac{\left\langle E_{L-1,q}, g_L + \frac{\|g_L\|^2}{\gamma} E_{L-1,q}\right\rangle^2 \|g_L + \frac{\|g_L\|^2}{\gamma} E_{L-1,q}\|^2}{\left(\|g_L + \frac{\|g_L\|^2}{\gamma} E_{L-1,q}\|^2 + \frac{1}{\eta}\right)^2} \right) \tag{27}$$

$$= -\sum_{q=1}^{m} \frac{\left(\|g_L + \frac{\|g_L\|^2}{\gamma} E_{L-1,q}\|^2 + \frac{2}{\eta}\right) \left\langle E_{L-1,q}, g_L + \frac{\|g_L\|^2}{\gamma} E_{L-1,q}\right\rangle^2}{\left(\|g_L + \frac{\|g_L\|^2}{\gamma} E_{L-1,q}\|^2 + \frac{1}{\eta}\right)^2}$$

$$= -\sum_{q=1}^{m} \frac{\left\langle E_{L-1,q}, g_L + \frac{\|g_L\|^2}{\gamma} E_{L-1,q}\right\rangle^2}{\left(\|g_L + \frac{\|g_L\|^2}{\gamma} E_{L-1,q}\|^2 + \frac{1}{\eta}\right)^2 / \left(\|g_L + \frac{\|g_L\|^2}{\gamma} E_{L-1,q}\|^2 + \frac{2}{\eta}\right)} \leq 0.$$

The monotonically decreasing property of $\|E_L\|$ has been proven. From Eqs. (25)–(27):

 

$$||E_L||^2 - (r+\mu_L)||E_{L-1}||^2$$

$$=\sum_{q=1}^{m}\left(\left\langle E_{L-1,q}-\beta_L\left(g_L+\frac{||g_L||^2}{\gamma}E_{L-1,q}\right),E_{L-1,q}-\beta_L\left(g_L+\frac{||g_L||^2}{\gamma}E_{L-1,q}\right)\right\rangle-(r+\mu_L)\left\langle E_{L-1,q},E_{L-1,q}\right\rangle\right)$$

$$=\sum_{q=1}^{m}\left(\left(1-r-\mu_L\right)\left\langle E_{L-1,q},E_{L-1,q}\right\rangle-2\left\langle E_{L-1,q},\beta_L\left(g_L+\frac{||g_L||^2}{\gamma}E_{L-1,q}\right)\right\rangle+\left\langle \beta_L\left(g_L+\frac{||g_L||^2}{\gamma}E_{L-1,q}\right),\beta_L\left(g_L+\frac{||g_L||^2}{\gamma}E_{L-1,q}\right)\right\rangle\right)$$

$$=\left(1-r-\mu_L\right)||E_{L-1}||^2-\sum_{q=1}^{m}\left(-2\frac{\left\langle E_{L-1,q},\beta_L\left(g_L+\frac{||g_L||^2}{\gamma}E_{L-1,q}\right)\right\rangle}{||g_L+\frac{||g_L||^2}{\gamma}E_{L-1,q}||^2+\frac{1}{\eta}}+\frac{\left\langle E_{L-1,q},g_L+\frac{||g_L||^2}{\gamma}E_{L-1,q}\right\rangle^2||g_L+\frac{||g_L||^2}{\gamma}E_{L-1,q}||^2}{\left(||g_L+\frac{||g_L||^2}{\gamma}E_{L-1,q}||^2+\frac{1}{\eta}\right)^2}\right)$$
(28)

$$=\left(1-r-\mu_L\right)||E_{L-1}||^2-\sum_{q=1}^{m}\frac{\left(||g_L+\frac{||g_L||^2}{\gamma}E_{L-1,q}||^2+\frac{2}{\eta}\right)\left\langle E_{L-1,q},g_L+\frac{||g_L||^2}{\gamma}E_{L-1,q}\right\rangle^2}{\left(||g_L+\frac{||g_L||^2}{\gamma}E_{L-1,q}||^2+\frac{1}{\eta}\right)^2}$$

$$=\left(1-r-\mu_L\right)||E_{L-1}||^2-\sum_{q=1}^{m}\frac{\left\langle E_{L-1,q},g_L+\frac{||g_L||^2}{\gamma}E_{L-1,q}\right\rangle^2}{\left(||g_L+\frac{||g_L||^2}{\gamma}E_{L-1,q}||^2+\frac{1}{\eta}\right)^2\Big/\left(||g_L+\frac{||g_L||^2}{\gamma}E_{L-1,q}||^2+\frac{2}{\eta}\right)}.$$

According to Eq. (25):

$$||E_L||^2-(r+\mu_L)||E_{L-1}||^2\le 0.$$
(29)

Therefore:

$$||E_L||^2\le r||E_{L-1}||^2+\mu_L||E_{L-1}||^2.$$
(30)

Theorem 1 has given $\lim_{L\to+\infty}\mu_L=0$, which means $\lim_{L\to+\infty}\mu_L||E_{L-1}||^2=0$. Based on Eq. (30), $\lim_{L\to+\infty}||E_L||^2=0$. Therefore, $\lim_{L\to+\infty}||E_L||=0$.

*Remark 1. In theorem 1, the output weights are evaluated by Eq. (26) and kept fixed. This may cause a slow convergence rate. To cope with this problem, the output weights of all hidden neurons are updated by the least squares method after the new hidden node has been added. Let $[\beta_1^*,\beta_2^*,...,\beta_L^*]=\arg\min\frac{\eta}{2}||f-(G+\frac{\widetilde{G}}{\gamma}\circ E)\beta||^2+\frac{1}{2}||\beta||^2$, where $G=[g_1,g_2,...,g_L]$, $\widetilde{G}=[||g_1||^2,||g_2||^2,...,||g_L||^2]$, $E=[E_0^*,E_1^*,...,E_{L-1}^*]$, '$\circ$' denotes the Hadamard product (element-wise multiplication) and $E_L^*=f-\sum_{j=1}^{L}\beta_j^*\left(g_j+\frac{||g_L||^2}{\gamma}E_{j-1}^*\right)$.*

*The output weights are calculated by:*

$$[\beta_1^*,\beta_2^*,...,\beta_L^*]=\arg\min_{\beta}\frac{\eta}{2}||f-\left(G+\frac{\widetilde{G}}{\gamma}\circ E\right)\beta||^2+\frac{1}{2}||\beta||^2$$

$$=\left(\left(G+\frac{\widetilde{G}}{\gamma}\circ E\right)^T\left(G+\frac{\widetilde{G}}{\gamma}\circ E\right)+\frac{I}{\eta}\right)^{-1}\left(G+\frac{\widetilde{G}}{\gamma}\circ E\right)^T f.$$
(31)

*The output weights are recalculated in accordance with Eq. (31) as the newly added hidden neuron is generated to satisfy Eq. (25). The inequality constraint guarantees the universal approximation capability of RSCN. The process of proof is similar to theorem 1, so the detailed proof procedure is omitted.*

**Remark 2.** *In Eq. (22), the residual error $\left( \frac{||g_L||^2}{\gamma} E_{L-1} \right)$ is added into the regularization term. The reason why $\frac{||g_L||^2}{\gamma}$ is used instead of $\gamma$ is that the value of the residual error is equal to the output of training samples before adding hidden neurons into the network $(E_0 = T)$. So the residual error is relatively larger at the beginning of the construction process. It gradually decreases as the constructive process proceeds. Meanwhile, due to the randomness of SCN, $g_L$ is randomly generated under a set of inequality constraints. The scale factor $\frac{||g_L||^2}{\gamma}$ makes it possible for the feedback residual error $\left( \frac{||g_L||^2}{\gamma} E_{L-1} \right)$ to adjust dynamically in pace with the change of the hidden output $(g_L)$.*

## RSCN-INFO algorithm

INFO is a very competitive new optimization algorithm. In this section, INFO is applied to optimize the parameter $\lambda$, the contractive factor $r$, the regularization coefficient $\eta$, and the positive scale factor $\gamma$ for RSCN. The widely-used root mean square error (RMSE) is employed as the fitness function.

$$RMSE = \sqrt{\frac{1}{N} \sum_{i=1}^{N} \left[ \sum_{j=1}^{L} \beta_j g_j(w_j^T x_i + b_j) - t_i \right]^2}. \tag{32}$$

For convenience's sake, $\xi_{L,q}, q = 1, 2, ..., m$ is defined to describe the algorithm. The pseudo-code of RSCN-INFO is summarized in Algorithm 1.

$$\xi_{L,q} = \frac{\left\langle E_{L-1,q}, g_L + \frac{||g_L||^2}{\gamma} E_{L-1,q} \right\rangle^2}{\zeta} - (1 - r - \mu_L)||E_{L-1,q}||^2 \tag{33}$$

where

$$\zeta = \frac{\left( ||g_L + \frac{||g_L||^2}{\gamma} E_{L-1,q}||^2 + \frac{1}{\eta} \right)^2}{||g_L + \frac{||g_L||^2}{\gamma} E_{L-1,q}||^2 + \frac{2}{\eta}}. \tag{34}$$

## The calculation complexity

The computational complexity of the INFO algorithm depends on the size of the population $N_P$, the times of iterations $G_{max}$, and the dimensional search domain $D$. The complexity of INFO is $O(N_P \times G_{max} \times D)$. For SCN, assume that a set of datasets with $N$ inputs $X = \{x_1, x_2, ..., x_N\}$, and the maximum number of hidden layer neurons of SCN is $L_{max}$. The main cost of SCN is caused by computing Moore-Penrose pseudo-inverse $H_L^\dagger T$. A rough estimate of the computational complexity of $H_L^\dagger$ can be expressed as $O(NL_{max}^3 + N^2 L_{max}^2 + L_{max}^4)$ (*Li & Wang, 2017*). Note that the cost of $H_L^\dagger T$ is calculated by

---

**Algorithm 1** RSCN-INFO

Training dataset: $(x_i, t_i)_{i=1}^{N}$, $x_i \in R^d, t_i \in R^m$.

Parameters: the population size $N_P$, maximum number of generations $G_{max}$, dimensional search domain $D$, upper bounds $ub$ and lower bounds $lb$ of $\lambda$, $r$, $\eta$ and $\gamma$, maximum number of hidden layer neurons $L_{max}$, residual error threshold $\tau$, maximum number of random assignment $T_{max}$.

Output: $v_{best}, f_{best}$,

1: **STEP 1: Initialization**

2: Initialize $E_0 = T, \Omega, W := [\quad]$;

3: Produce an initial population $P^0 = \{v_1^0, v_2^0, ..., v_{N_P}^0\}$, where $v_i^0 = \{v_{i,\lambda}^0, v_{i,r}^0, v_{i,\eta}^0, v_{i,\gamma}^0\}$;

4: Calculate $f(v_i^0)$ by Eq. (32);

5: **STEP 2: Parameter optimization by INFO**

6: **for** $g = 1$ to $G_{max}$ **do**

7:     **for** $i = 1$ to $N_P$ **do**

8:         Randomly choose three vectors $(x_{a1}, x_{a2}, x_{a3})$, and calculate $w$ by Eqs. (7)–(9) and (12)–(14);

9:         Create two new vectors using Eq. (16);

10:         The two new vectors are combined by Eq. (17);

11:         Execute local search using Eqs. (18)–(20);

12:         Update the Vector $v_i^g = \{v_{i,\lambda}^g, v_{i,r}^g, v_{i,\eta}^g, v_{i,\gamma}^g\}$;

13:         **while** $L \le L_{max}$ AND $||E_0|| \ge \tau$ **do**

14: **STEP 3: Hidden parameter configuration (Step 15–28)**

15:             **for** $\lambda = v_{i,\lambda}^g$, $r = v_{i,r}^g$, $\eta = v_{i,\eta}^g$ and $\gamma = v_{i,\gamma}^g$ **do**

16:                 **for** $k = 1, 2, ..., T_{max}$ **do**

17:                     Randomly select $w_L$ and $b_L$ from $\lambda^d$ and $\lambda$;

18:                     Compute $g_L$ and $\xi_{L,q}$ by Eqs. (33) and (34)

19:                     **if** $w_L^*$ and $b_L^*$ satisfy constraint inequality **then**

20:                         Save the random parameters in $W$, $\xi_L$ in $\Omega$;

21:                     **end if**

22:                 **end for**

23:                 **if** $W$ is not empty **then**

24:                     Choose $w_L^*$ and $b_L^*$ corresponding to the maximize $\xi_L$ in $\Omega$, set $G = [g_1^*, g_2^*, ..., g_L^*]$ and $E = [E_0^*, E_1^*, ..., E_{L-1}^*]$;

25:                     Break (go to Step 30);

26:                 **else**    Return to Step 7;

27:                 **end if**

28:             **end for**

29: **STEP 4: Output weight determination (Step 30–32)**

30:             Calculate $[\beta_1^*, \beta_2^*, ..., \beta_L^*] = \left((G + \frac{\widetilde{G}}{\gamma} \circ E)^T(G + \frac{\widetilde{G}}{\gamma} \circ E) + \frac{I}{\eta}\right)^{-1} (G + \frac{\widetilde{G}}{\gamma} \circ E)^T f$;

31:             Calculate $E_L = T - (G + \frac{\widetilde{G}}{\gamma} \circ E)\beta^*$;

32:             Renew $E_0 := E_L, L := L + 1$;

33:     **end while**

---

(Continued)

| Algorithm 1 (continued) |
|---|

34:          Determine $\beta^*$, $w^*$ and $b^*$;

35:          Calculate the fitness function $f(v_i^g)$ according to Eq. (32).

36:          **if** $f(v_i^g) < f_{best}$ **then** $f_{best} = f(v_i^g)$, $v_{best} = v_i^g$

37:          **end if**

38:      **end for**

39:      Update the best vector $v_{best} = \{v_{best,\lambda}, v_{best,r}, v_{best,\eta}, v_{best,\gamma}\}$;

40: **end for**

41: Return $v_{best}, f_{best}$

the widely-used singular value decomposition. Hence, the total complexity of RSCN-INFO is $O(N_P \times G_{max} \times D \times (NL_{max}^3 + N^2 L_{max}^2 + L_{max}^4))$.

## EXPERIMENTS

The effectiveness of RSCN-INFO was evaluated on a function approximation and three benchmark datasets from the Knowledge Extraction based on Evolutionary Learning (KEEL, http://www.keel.es) dataset repository supported by the Spanish Ministry of Science and Technology. The approximation function is a conventional high nonlinear compound function that is widely used to evaluate randomized neural networks. The KEEL dataset contains classification, regression, unsupervised, and time series datasets. To verify the effectiveness of RSCN-INFO, it was compared with classical IRVFL, SCN (*Wang & Li, 2017b*), RSCN (*Wang et al., 2021*), and DASCN-II (*Wang et al., 2020*). All the experiments were implemented with MATLAB R2019b on a PC with AMD Ryzen 7 3.20 GHz CPU, NVIDIA GeForce MX450 GPU, and 16 GB RAM.

### Function approximation

Let the real-valued function $f(x)$ be defined as follows (*Tyukin & Prokhorov, 2009*):

$$y = 0.2e^{-(10x-4)^2} + 0.5e^{-(80x-40)^2} + 0.3e^{-(80x-20)^2}, x \in [0,1]. \tag{35}$$

We randomly generated 1,000 training samples and 300 test samples from the uniform distribution, and a regularly spaced grid over (0,1). Figure 2 compares the function approximation performance of RSCN-INFO with IRVFL, SCN, RSCN, and DASCN-II. Since our proposed RSCN-INFO could achieve reliable and accurate performance with the lower number of hidden neurons, the value of $L_{max}$ was set to 25. In the simulations, the value of RMSE remained virtually unchanged when the widely used setting $[-1,1]$ was set for IRVFL. So the scope of random parameters for IRVFL was set as $(-250, 250)$. For SCN, the value of $T_{max}$ was set to 100, $\lambda$ and $r$ were selected from the set $\{100 : 1 : 200\}$ and $\{0.9, 0.99, 0.999, 0.9999, 0.99999, 0.999999\}$. In RSCN-INFO, the population size $N_p$ and the maximum generations were set to 30 and 10, respectively. The lower bounds and upper bounds of $\lambda$, $r$, $\eta$, and $\gamma$ were set to $[100, 200]$, $[0.9, 0.999999]$, $[0, 2^{40}]$, and $[10^5, 10^9]$. As

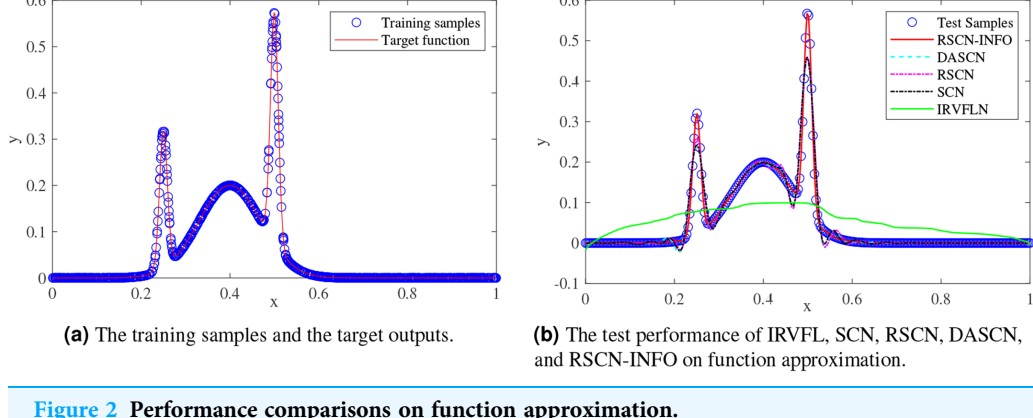

**(a)** The training samples and the target outputs.

**(b)** The test performance of IRVFL, SCN, RSCN, DASCN, and RSCN-INFO on function approximation.

**Figure 2 Performance comparisons on function approximation.**

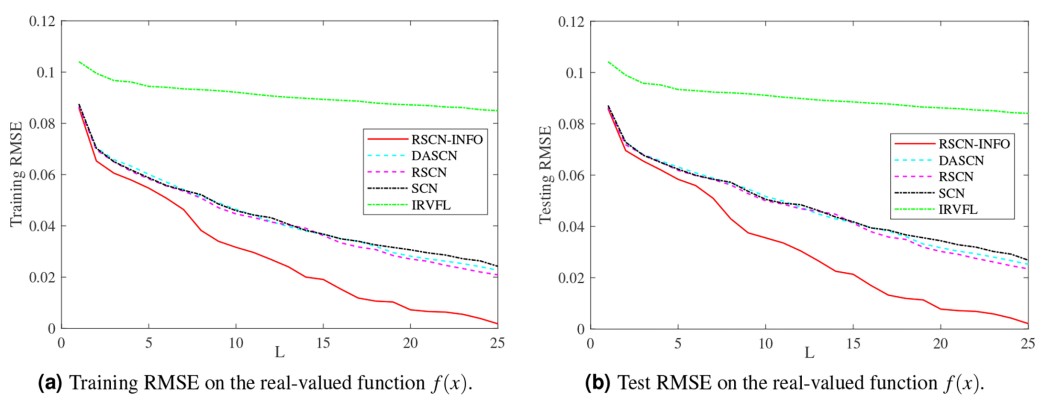

**(a)** Training RMSE on the real-valued function $f(x)$.

**(b)** Test RMSE on the real-valued function $f(x)$.

**Figure 3 Average RMSE on the real-valued function $f(x)(L_{max} = 25)$.**

seen in Fig. 2B, the IRVFL showed far worse performance than that of SCNs, while the performance of our proposed RSCN-INFO was the best.

Figures 3 and 4 display the training and test results of the real-value function with 25 and 50 hidden nodes, respectively. The average training RMSE was obtained from 20 independent experiments. For IRVFL, Figs. 3 and 4 clearly show that the training RMSE was unacceptable. Furthermore, the convergence rate of RSCN-INFO is faster than that of SCN, RSCN, and DASCN-II, which verifies the efficiency of RSCN-INFO. In addition, Table 1 reports the average RMSE and standard deviation results of different models. It is evident that RSCN-INFO achieved more favorable results than the other algorithms.

## Benchmark datasets

Three real-world benchmark datasets for regression from KEEL were employed as experimental datasets. Specifications of these datasets are given in Table 2.

Figures 5–7 and Tables 3–5 depict the average training and test results on these benchmark datasets. Each test's statistical results for the 20 run times and the average value of RMSE were selected to evaluate the performance of the different algorithms. The IRVFL could not reach the preset tolerance threshold, so it was omitted here. In this case, the

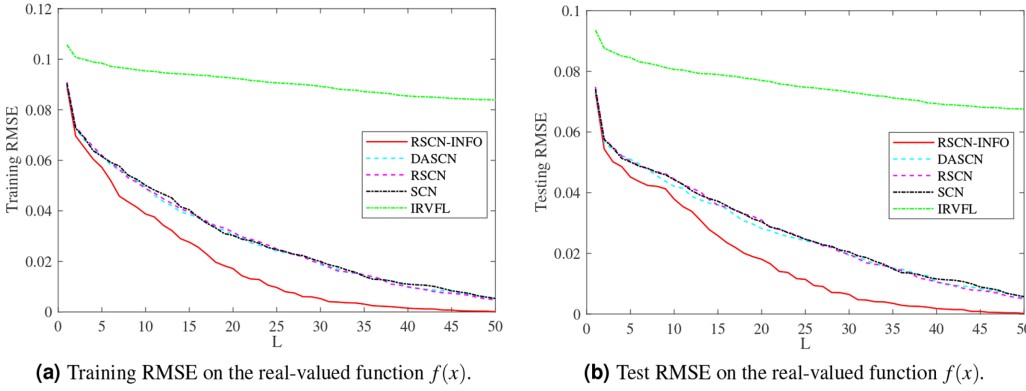

**(a)** Training RMSE on the real-valued function $f(x)$.

**(b)** Test RMSE on the real-valued function $f(x)$.

**Figure 4** Average RMSE on the real-valued function $f(x)(L_{max} = 50)$.

**Table 1** Performance comparisons of different methods on function approximation.

| Methods | Training results | | Test results | |
|---|---|---|---|---|
| | $L = 25$ | $L = 50$ | $L = 25$ | $L = 50$ |
| IRVFL | $0.08493 \pm 0.00548$ | $0.08389 \pm 0.00426$ | $0.08405 \pm 0.00524$ | $0.06756 \pm 0.00559$ |
| SCN | $0.02421 \pm 0.00479$ | $0.00535 \pm 0.00342$ | $0.02681 \pm 0.00535$ | $0.00570 \pm 0.00364$ |
| RSCN | $0.02090 \pm 0.00448$ | $0.00477 \pm 0.00222$ | $0.02344 \pm 0.00499$ | $0.00506 \pm 0.00252$ |
| DASCN-II | $0.02267 \pm 0.00345$ | $0.00512 \pm 0.00268$ | $0.02523 \pm 0.00368$ | $0.00553 \pm 0.00285$ |
| RSCN-INFO | $0.00179 \pm 0.00025$ | $0.00015 \pm 0.00008$ | $0.00209 \pm 0.00031$ | $0.00016 \pm 0.00009$ |

**Table 2** Specifications of three benchmark regression datasets.

| Dataset | Attributes | | Instances |
|---|---|---|---|
| | Features | Output | |
| Concrete | 8 | 1 | 1,030 |
| Compactiv | 21 | 1 | 8,192 |
| Pole | 26 | 1 | 14,998 |

scope of random parameters $\lambda$ in SCN was selected from the set $\{1 : 0.1 : 5\}$ and the lower and upper bounds of $\lambda$ in RSCN-INFO were set to $[1, 5]$. All the other parameters were set the same as the function approximation.

Figure 5 shows similar performance between RSCN-INFO and the competitor algorithms on concrete. The reason for this phenomenon is that the concrete dataset contained smaller features and instances. For the compactiv and pole datasets, Figs. 6 and 7 clearly show that RSCN-INFO can achieve lower RMSE in terms of both training and test results. Intuitively and obviously, the RMSE of RSCN was used as the fitness function of INFO. In essence, INFO explored a global optimum solution that minimizes the fitness function in a four-dimensional search domain $(\lambda, r, \gamma, \eta)$ over several successive generations.

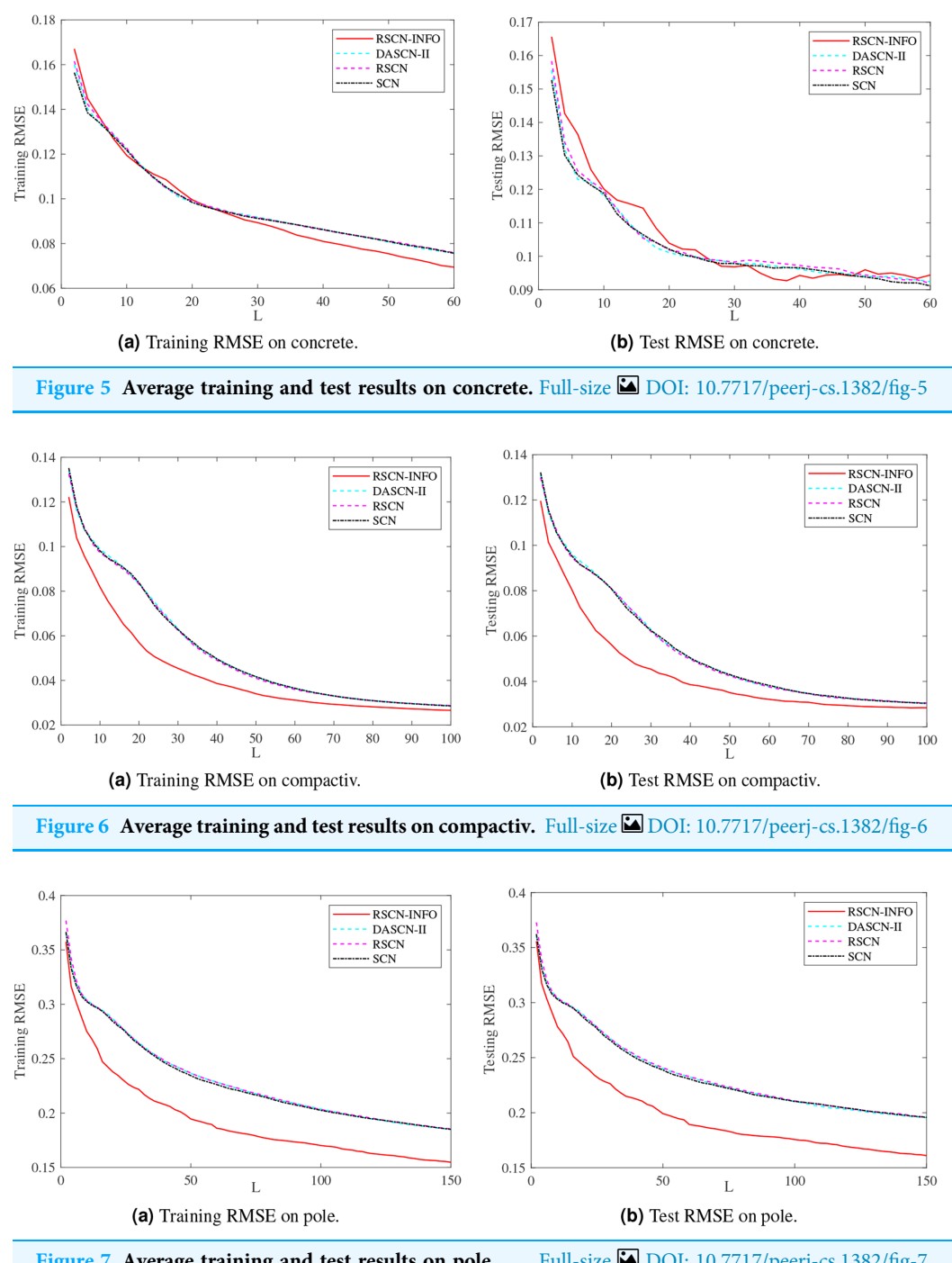

**(a)** Training RMSE on concrete.

**(b)** Test RMSE on concrete.

**Figure 5 Average training and test results on concrete.**

**(a)** Training RMSE on compactiv.

**(b)** Test RMSE on compactiv.

**Figure 6 Average training and test results on compactiv.**

**(a)** Training RMSE on pole.

**(b)** Test RMSE on pole.

**Figure 7 Average training and test results on pole.**

To verify the effectiveness of RSCN-INFO, Table 6 lists the computational time among SCN, RSCN, DASCN-II, and RSCN-INFO on benchmark datasets. We found that the training time of RSCN-INFO was significantly shorter than the other methods on the three benchmark datasets. Table 6 indicates that RSCN-INFO which employed optimized parameters and could achieve better efficiency. It should be noted that we did not take the parameter optimization process into account in this experiment. The optimization process

**Table 3 Performance comparisons of different methods on concrete.**

| Methods | Training, Test results | | | | |
|---|---|---|---|---|---|
| | L = 20 | L = 30 | L = 40 | L = 50 | L = 60 |
| SCN | 0.09842, 0.10212 | 0.09122, 0.09786 | 0.08626, 0.09655 | 0.08098, 0.09381 | 0.07566, 0.09114 |
| RSCN | 0.09912, 0.10187 | 0.09156, 0.09831 | 0.08613, 0.09724 | 0.08113, 0.09438 | 0.07594, 0.09190 |
| DASCN-II | 0.09840, 0.10109 | 0.09175, 0.09769 | 0.08626, 0.09613 | 0.08066, 0.09405 | 0.07566, 0.09239 |
| RSCN-INFO | 0.09957, 0.10391 | 0.08933, 0.09684 | 0.08097, 0.09427 | 0.07545, 0.09596 | 0.06944, 0.09439 |

**Table 4 Performance comparisons of different methods on compactiv.**

| Methods | Training, Test results | | | | |
|---|---|---|---|---|---|
| | L = 20 | L = 40 | L = 60 | L = 80 | L = 100 |
| SCN | 0.08380, 0.08076 | 0.04963, 0.05035 | 0.03646, 0.03823 | 0.03090, 0.03264 | 0.02855, 0.03032 |
| RSCN | 0.08317, 0.08091 | 0.04902, 0.04981 | 0.03595, 0.03752 | 0.03093, 0.03243 | 0.02872, 0.03049 |
| DASCN-II | 0.08345, 0.08081 | 0.04972, 0.05003 | 0.03638, 0.03781 | 0.03096, 0.03263 | 0.02858, 0.03028 |
| RSCN-INFO | 0.05695, 0.05613 | 0.03863, 0.03856 | 0.03117, 0.03211 | 0.02815, 0.02932 | 0.02658, 0.02839 |

**Table 5 Performance comparisons of different methods on pole.**

| Methods | Training, Test results | | | | |
|---|---|---|---|---|---|
| | L = 30 | L = 60 | L = 90 | L = 120 | L = 150 |
| SCN | 0.26345, 0.26582 | 0.22619, 0.23069 | 0.20756, 0.21426 | 0.19464, 0.20406 | 0.18507, 0.19572 |
| RSCN | 0.26411, 0.26746 | 0.22831, 0.23278 | 0.20871, 0.21545 | 0.19524, 0.20409 | 0.18542, 0.19602 |
| DASCN-II | 0.26462, 0.26741 | 0.22856, 0.23264 | 0.20895, 0.21497 | 0.1945, 0.20299 | 0.18478, 0.19538 |
| RSCN-INFO | 0.22182, 0.22629 | 0.18604, 0.18925 | 0.17349, 0.17836 | 0.16272, 0.16904 | 0.15479, 0.16099 |

**Table 6 The computational time of different algorithms on benchmark datasets.**

| Datasets | Algorithms | Error tolerance $\tau$ | Training time (Mean ± STD) |
|---|---|---|---|
| Concrete | SCN | 0.08 | 0.1977 ± 0.00379 |
| | RSCN | | 0.2083 ± 0.03008 |
| | DASCN-II | | 0.2275 ± 0.02442 |
| | RSCN-INFO | | 0.1021 ± 0.01095 |
| Compactiv | SCN | 0.03 | 2.1405 ± 0.25356 |
| | RSCN | | 2.1773 ± 0.16212 |
| | DASCN-II | | 2.5231 ± 0.22985 |
| | RSCN-INFO | | 1.2148 ± 0.22618 |
| Pole | SCN | 0.20 | 6.5387 ± 0.94180 |
| | RSCN | | 6.2857 ± 0.68270 |
| | DASCN-II | | 7.9776 ± 1.52610 |
| | RSCN-INFO | | 1.7529 ± 0.28892 |

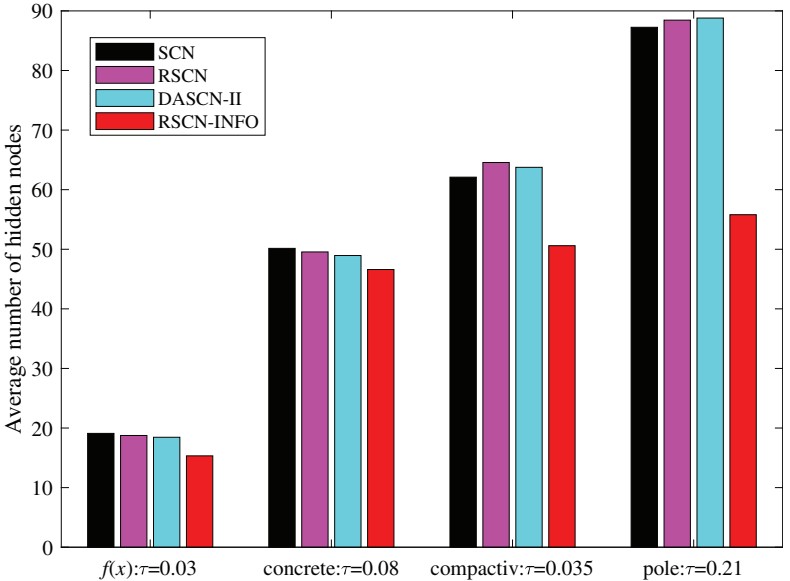

**Figure 8** **Average number of hidden nodes on $f(x)$ and benchmark datasets.**

may consume additional time. However, the improvement of regression accuracy and network structure may be worth the time that is spent on the parameter optimization.

To further illustrate the network compactness of RSCN-INFO, we investigated how many hidden nodes were required to meet a preset error tolerance. As shown in Fig. 8, RSCN-INFO requires fewer hidden nodes compared with other methods. It can be deduced that given a preset $\tau$, RSCN-INFO can reach the error tolerance using fewer hidden neurons. This is due to RSCN-INFO using optimized parameters that can achieve a higher residual error reduction. Therefore, the network structure is more compact. It should be pointed out that DASCN-II can also construct a relatively compact SCN. However, the tunable value $\gamma$ in DASCN-II is a fixed value. It is selected empirically and difficult to adjust. Moreover, an inappropriate $\gamma$ will seriously affect the accuracy of the model.

In classical SCN and its various variants, $\lambda$ tends to set a relatively larger value in complex problems. The parameter $r$ is unfixed and set based on an increasing sequence from 0.9 to 1. The other parameters are selected empirically in connection with practical applications. Therefore, the conclusion may be drawn that RSCN-INFO is not only helpful in adaptively selecting parameters of SCN, but also beneficial for constructing a compact network.

## CONCLUSION

This article developed a new regularized SCN based on the INFO optimization algorithm, named RSCN-INFO. On one hand, the added regularization term combines the ridge method with the residual error feedback, contributing to the balance of the structural (output weights) and empirical (network residual error) losses of SCN. On the other hand, RMSE was selected as the fitness function of INFO to assist SCN to locate up-and-coming

areas in multi-dimensional search space. A higher residual error decreasing rate is impacted by the parameter selection of RSCN-INFO. The experimental results on a function approximation and three benchmark regression datasets from KEEL indicated that the proposed RSCN-INFO algorithm exhibits considerable advantages in parameter optimization and network structure compactness compared with other algorithms.

In almost all practical modeling tasks, the presence of noise and outliers is inevitable. This optimization strategy will accelerate the degradation of the learning performance of SCN that are subjected to noise or outliers. The robust skills used to weaken the negative influences of noise and outliers will be further discussed.

### Funding
This work is supported by the National Natural Science Foundations of China (No. 62163007 and No. 62166005). The funders had no role in study design, data collection and analysis, decision to publish, or preparation of the manuscript.

### Grant Disclosures
The following grant information was disclosed by the authors:
National Natural Science Foundations of China: 62163007 and 62166005.

### Competing Interests
The authors declare that they have no competing interests.

### Author Contributions
- Yang Wang conceived and designed the experiments, analyzed the data, performed the computation work, prepared figures and/or tables, authored or reviewed drafts of the article, and approved the final draft.
- Tao Zhou conceived and designed the experiments, performed the experiments, performed the computation work, prepared figures and/or tables, authored or reviewed drafts of the article, and approved the final draft.
- Guanci Yang performed the experiments, authored or reviewed drafts of the article, and approved the final draft.
- Chenglong Zhang analyzed the data, authored or reviewed drafts of the article, and approved the final draft.
- Shaobo Li conceived and designed the experiments, authored or reviewed drafts of the article, and approved the final draft.

### Data Availability
The raw training, test datasets, three benchmark datasets (concrete, compactiv, pole), and code are available in the Supplemental Files.

## Supplemental Information

Supplemental information for this article can be found online at http://dx.doi.org/10.7717/peerj-cs.1382#supplemental-information.

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
