# Peer review of "A regularized stochastic configuration network based on weighted mean of vectors for regression"

_PeerJ Computer Science, doi:10.7717/peerj-cs.1382_

## Round 0.1 · original submission · Major Revisions

The paper contains publishable contents in terms of technical contribution. However, the writing needs to be further improved as the readability of current version affects the evaluation of the proposed method performance. Therefore, a major revision is needed to address all the comments from reviewers and a proofreading is highly recommended.

Reviewer 1 ·

Basic reporting

The manuscript addresses an important practical and theoretical challenge of developing efficient learning algorithms with proven generalisation capabilities. The Authors propose to combine classical SCNs with evolutionary optimisation to increase the efficiency of constructing regularised SCNs. In my opinion these results are important and new and deserve to be published. The manuscript is well-written and presented. I have only few minor comments, mostly around notation.

1. In or before eq (2), it would be great to define the meaning of <,> (I suppose that this is an inner product but it wasn't defined as such). It may be worthwhile to define domains of the functions f and g too.

2. In eq (20), please define h(x_i). Consider adding that \eta is non-negative

Experimental design

All experiments are clear

Validity of the findings

The results are clear and appear to be correct. Conclusions are adequate and all data are provided and sound.

Additional comments

No further comments

Reviewer 2 ·

Basic reporting

A new method of RSCN-INFO is proposed in this paper for optimizing parameter settings of the SCN method. The article structure is reasonable, and literature references,experiment result are sufficient. The definitions of theorems are clear.

The full-text language should be refined to improve the logic and causality of expression。For example, shallow neural networks is highlighted in abstract, while in the introduction section the neural networks are described.
1. “Stochastic configuration networks (SCN) have shown tremendous potential in building shallow neural networks under a supervisory mechanism to constrain the random assignment of hidden layer parameters.”, is it a precision description?
2.“the performance of SCN is frequently impacted by the parameter settings of the model”, it would be better to specify the major performance.
3. In Line 41, what is this problem, please express yourself clearly.
4. Check the equation (1) and (2), are they elements or vectors.
5. Check the pseudo-code carefully in page 8, such as the code in line 30.

Experimental design

The experiment and comparison test are convincing and sufficient. However, some improvement can be made,
In page9, some extra sentences, such as universality, should be added to describe briefly the background of the function and KEEL dataset.

Validity of the findings

The comparison of test results show that the RSCN model with paramenter settings and structure optimized by INFO possess superior performance, such as fast convergence, a more compact network structure.

---

## Round 0.2 · accepted · Accept

The manuscript is ready for publication.

Reviewer 1 ·

Basic reporting

All my previous concerns have been fully addressed in the revision.

Experimental design

n/a

Validity of the findings

No detectable errors

Additional comments

None